# Human blindsight is mediated by an intact geniculo-extrastriate pathway

Sara Ajina[1,2], Franco Pestilli[3], Ariel Rokem[4,5], Christopher Kennard[2], Holly Bridge[1,2]*

[1]Oxford Centre for Functional MRI of the Brain, University of Oxford, Oxford, United Kingdom; [2]Nuffield Department of Clinical Neurosciences, University of Oxford, Oxford, United Kingdom; [3]Department of Psychological and Brain Sciences, Programs in Neuroscience and Cognitive Science, Indiana University Network Science Institute, Indiana University, Bloomington, United States; [4]Department of Psychology, Stanford University, Stanford, United States; [5]eScience Institute, University of Washington, Seattle, United States

**Abstract** Although damage to the primary visual cortex (V1) causes hemianopia, many patients retain some residual vision; known as blindsight. We show that blindsight may be facilitated by an intact white-matter pathway between the lateral geniculate nucleus and motion area hMT+. Visual psychophysics, diffusion-weighted magnetic resonance imaging and fibre tractography were applied in 17 patients with V1 damage acquired during adulthood and 9 age-matched controls. Individuals with V1 damage were subdivided into blindsight positive (preserved residual vision) and negative (no residual vision) according to psychophysical performance. All blindsight positive individuals showed intact geniculo-hMT+ pathways, while this pathway was significantly impaired or not measurable in blindsight negative individuals. Two white matter pathways previously implicated in blindsight: (i) superior colliculus to hMT+ and (ii) between hMT+ in each hemisphere were not consistently present in blindsight positive cases. Understanding the visual pathways crucial for residual vision may direct future rehabilitation strategies for hemianopia patients.

*For correspondence:
holly.bridge@ndcn.ox.ac.uk

## Introduction

Following damage to the primary visual cortex (V1) patients experience homonymous hemianopia, in which vision on one side of the visual field is lost. However, in spite of this cortical blindness, some patients are still able to ascertain information about visual stimulation within the blind area; this is called blindsight. Over the past 30 years, several visual pathways have been proposed to underlie this residual vision, but the relative role of these pathways and the neurobiological bases for blindsight remains unknown (see *Cowey, 2010* for review).

Diffusion-weighted magnetic resonance imaging (dMRI) combined with tractography offers a practical and non-invasive method for estimating large-scale white matter tracts and studying their microstructural properties in living humans (*Johansen-Berg, 2010*; *Catani et al., 2012*; *Jones et al., 2013*). The method provides a unique approach to investigate how white matter properties relate to visual behaviour in blindsight.

Using dMRI in a number of individual patients, intact ipsilateral white matter connecting lateral geniculate nucleus (LGN) and extrastriate cortex, specifically area hMT+, has been proposed as a candidate circuit that could support blindsight (*de Gelder et al., 2008*; *Bridge et al., 2010*). In agreement with this proposal, the macaque LGN can support residual visual processing after V1 lesion (*Schmid et al., 2010*). Two alternative proposals suggest blindsight results either from visual plasticity, for example to strengthen interhemispheric white matter in humans (*Leh et al., 2006*; *Bridge et al., 2008*;

**eLife digest** Visual information from our eyes projects to a region at the back of the brain called the primary visual cortex, which is where the information is processed to allow us to see the world around us. If a person suffers a stroke that affects this primary visual cortex, he or she can become blind on one side. However, some people can still detect images within this 'blind' area, even if they are not consciously aware of it. This phenomenon is known as 'blindsight', but it remains unclear which pathways and structures in the brain might allow this information to be detected.

Ajina et al. have now examined the brains of a large group of patients with damage to the visual cortex. The results for the patients with blindsight were compared to those without, and to a group of sighted control participants. This analysis identified a pathway that seems to underlie blindsight. This pathway (which runs between an area of the brain called the lateral geniculate nucleus and another called the motion area hMT+) was present in all patients with blindsight, but was missing or disrupted in those patients without blindsight.

Ajina et al. then examined other pathways that had previously been suggested to support blindsight and revealed that they were unlikely to do so. This is because the suggested connections were not identifiable in all patients with blindsight, and were often intact in those patients without blindsight.

So far, this work has addressed the structure of the pathways rather than their activity. Future work will attempt to determine whether it is possible to strengthen such pathways to improve visual ability.

*Tamietto et al., 2012*) or intact connections to hMT+ from the superior colliculus and pulvinar, demonstrated in the macaque (*Warner et al., 2010*, *2015*). The superior colliculus has also been implicated in human residual vision after V1 damage, particularly for indirect blindsight and saccadic localisation (*Mohler and Wurtz, 1977*; *Leh et al., 2006*; *Kato et al., 2011*). To date the necessary circuitry supporting preserved vision after V1 damage in humans has not been identified.

The present study investigated visual white matter tracts in the largest group of patients measured to date with chronic unilateral V1 damage sustained in adulthood (n = 17, see *Supplementary file 1* for clinical and demographic details) and healthy age-matched controls (n = 9). The large subject group enabled the division of patients into those demonstrating blindsight, and those who did not. Three pathways were selected (1) ipsilateral connections between the LGN and hMT+, (2) ipsilateral tracts between the superior colliculus and hMT+, and (3) interhemispheric tracts between hMT+ bilaterally. We evaluated the ability to identify these tracts in all individuals and characterised their anatomy and white matter properties.

The preservation or destruction of the geniculate-hMT+ tract predicted presence or absence of blindsight respectively. More specifically, the geniculate-hMT+ tract was reliably identified in all blindsight positive patients, but was impossible to track or showed considerably impaired white matter microstructure in all blindsight negative individuals. In contrast, the two alternative candidate tracts showed variable predominance in both patient groups and therefore seem unlikely to underlie blindsight function.

## Results

### Behavioural measurements of blindsight

Blindsight was determined according to performance on a high salience 2-AFC temporal detection paradigm presented within the blind region of the visual field (*Figure 1A*). Patients detected the interval in which the target appeared (*Figure 1B*) and were classified as 'blindsight positive' if average performance or performance for stimuli of 100% contrast was significantly above chance (*Figure 1C*; *Ajina et al., 2015b*). Based on these criteria, 12 were classified as 'blindsight positive' and this relatively sensitive binary measure allowed us to be confident that patients labelled as 'blindsight negative' (n = 5) showed no residual visual function. No patients could describe the stimulus in their blind field, although the degree of awareness varied from a complete absence of awareness to an appreciation of motion at times in the minority of cases.

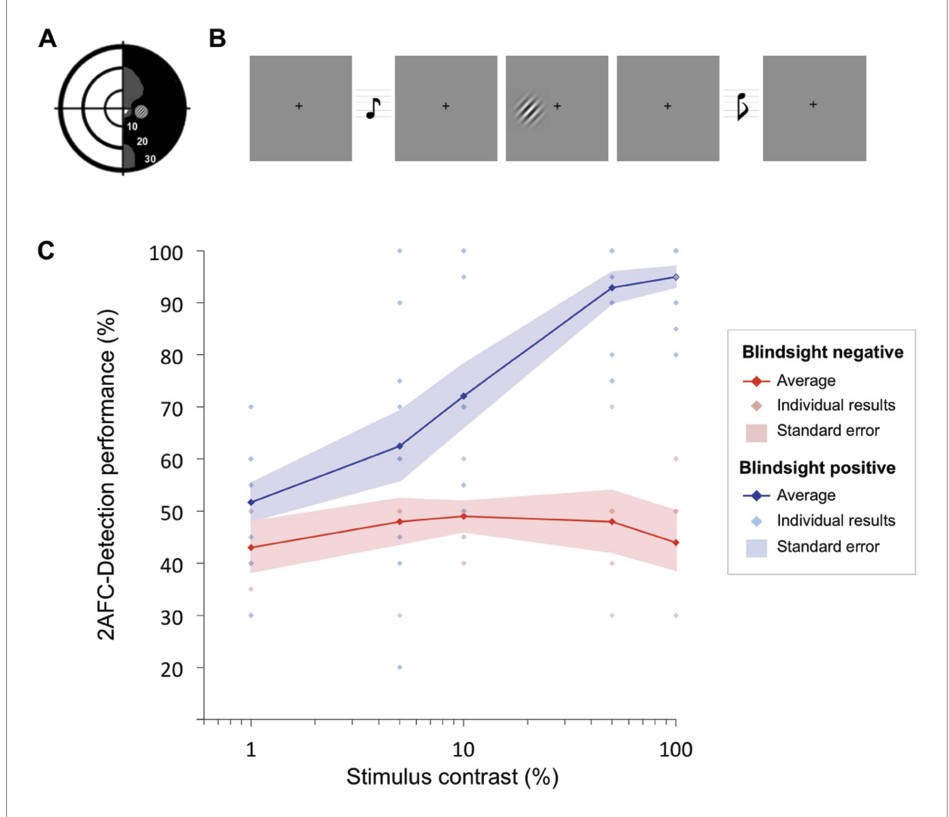

**Figure 1**. Psychophysics protocol and results. (**A**) Example Humphrey visual field deficit drawn schematically, with the location of the target stimulus superimposed. Dense visual field loss is shown in black (<0.5%) and partial loss in grey (<2%). (**B**) Illustration of the 2AFC-temporal detection procedure. Participants fixated on a central cross, with the onset of each 1500ms interval alerted by a low (interval 1) or high pitch (interval 2) tone. The stimulus could appear in either interval, for a period of 500 ms. At the end of the trial, participants were instructed to decide in which interval the stimulus appeared. (**C**) Detection performance with increasing stimulus contrast, shown separately for blindsight positive (blue) and blindsight negative (red) patients. Individual results are also plotted for each patient. Chance level is 50%.

Classification of participants as either 'blindsight positive' or 'blindsight negative' was further validated using cross-validation. Two different classification algorithms were applied to participants' performance across all contrast levels and they were compared to classification based only on performance at 100% contrast. The first algorithm was k-nearest-neighbours: it classified participants based on the labels ('blindsight positive' or 'blindsight negative') assigned to the majority of the 5 participants with behavior most similar to theirs, based only on the performance of these 5 participants at 100% contrast. The second algorithm was a Gaussian mixture model classification: centroids of two Gaussian distributions were used to fit the data without labels (i.e. with no knowledge of whether *any* of the participants were classified as 'blindsight positive' or 'blindsight negative' based on performance at 100% contrast). Each participant was then assigned to one of the two classes based on their similarity to the centroid of each of these distributions. Both classification algorithms agree with the distinction based on performance in 100% contrast in all cases.

## White matter tracts between LGN and hMT+ are demonstrable in the majority of patients

All 12 blindsight positive patients and the 9 age-matched controls were found to have ipsilateral, uncrossed tracts between the LGN and hMT+. We combined High-Angular Resolution Diffusion-weighted magnetic resonance Imaging (HARDI; 60 diffusion directions, b-value = 1500) and modern

probabilistic tractography (*Tournier et al., 2012*; see 'Materials and methods' for more details) to track between different pairs of regions of interest (ROIs) with a fixed number of fascicles, or steamlines (target 10,000, max generated 1,000,000). We counted the number of fascicles between each of several ROI pairs in each brain. The precise number of fascicles is an arbitrary value, dependent on many interacting tracking parameters and properties of the measured diffusion data (see *Pestilli et al., 2014* and 'Discussion' for more details). Here, we used fascicle count as an indirect measure of the difficulty of tracking a white matter pathway. To further standardise the measure, we used an anatomical standardisation method to eliminate outlier fascicles from counts while constraining the number to a conservative lower bound within each individual brain. This was achieved by removing outlier fascicles defined as those more than about 2.5 standard deviations away from or longer than each core tract (2.6 and 2.8 SD respectively; see *Allen et al., 2015*; *Pestilli et al., 2014*; *Yeatman et al., 2012* and 'Materials and methods' for details). This process generated a core tract-bundle containing a conservative 25% $\pm$ 8% of the original number of fascicles in each subject (see *Supplementary file 2* for original numbers).

The number of fascicles measured was of similar magnitude in control and blindsight positive individuals. All tracts were reliably measured in both hemispheres, including the hemisphere with V1 damage in blindsight positive patients (see *Table 1*). As expected (*Jones et al., 2013*; *Pestilli et al., 2014*), even after cleaning, there was considerable variation in fascicle numbers between participants and, in some cases, between hemispheres although this variability was similar for controls (range = 19–653) and patients (range = 17–635). In blindsight negative patients it was possible to track a pathway between the LGN and hMT+ in the damaged hemisphere of 4/5 patients (we failed to identify the tract in PN4), with a similar number of fascicles to blindsight positive patients (*Figure 2A*, *Table 1*). However, all of these patients showed considerable abnormality in the microstructure of these tracts compared to their intact hemisphere or control participants, highlighting the importance of considering white matter microstructure in patient tractography studies. *Figure 2A* shows examples of the anatomical trajectory of these identified pathways for participants from the three groups.

## Different white matter microstructure in the geniculate-hMT+ tract between blindsight positive and negative patients

Fractional anisotropy (FA) and mean diffusivity (MD) are commonly used diffusion MRI indices, representing tissue microstructure in situations of neuronal damage (*Werring et al., 2000*; *Jones et al., 2013*). These measures are quite sensitive to a number of tissue properties, such as axonal ordering, axonal packing density, degree of myelination, membrane permeability, without being very specific to any one of them. This can result in difficulties related to interpretation. FA is derived from the relationship between the amounts of free water anisotropic diffusion in a single (principal) direction, relative to all other directions (*Basser, 1995*; *Jones et al., 2013*). Decreases in FA have been associated with impaired tissue microstructure. MD is a measure of the total mean diffusion magnitude in all directions in a voxel, and its value also reflects a complex relationship with tissue microstructure. In general, white matter tissue damage has been associated with an increase in MD (*Jones et al., 2013*). We used an advanced tract-anatomy informed analysis (*Allen et al., 2015*; *Yeatman et al., 2012*; see 'Materials and methods' for more details) and measured FA and MD along the length of each individual tract. We then computed the mean FA and MD measures along each tract using the core portion of the pathway to eliminate artifactual measurements due to potential partial voluming with grey matter and scar tissue. Mean FA and MD for each core tract were averaged for each participant to generate separate measures for the ipsilesional and intact hemispheres. Mean FA, calculated across all blindsight positive patients and collapsed along the whole geniculate-hMT+ pathway, was 0.43 $\pm$ 0.05 (mean $\pm$ s.d.) in the damaged hemisphere and 0.49 $\pm$ 0.05 in the intact hemisphere, corresponding to a laterality of 13.7% (see *Figure 2B* and 'Materials and methods' for details). Laterality, representing the relative difference in diffusivity for equivalent tracts in opposite hemispheres, was slightly more prominent over the early-mid portions of the pathway. In blindsight negative patients (*Figure 2B*, middle column), the microstructure of ipsilesional tracts was particularly abnormal. Mean FA was 0.35 $\pm$ 0.1 (mean $\pm$ s.d.) on the ipsilesional side, vs 0.47 $\pm$ 0.03 in the intact hemisphere (laterality = 34.7%). In comparison, control participants (*Figure 2B*, right column) show a left-right laterality of 3.3% for FA (range = 0.3–0.66. Mean FA = 0.51 $\pm$ 0.03 left, 0.49 $\pm$ 0.03 right

**Table 1**. Number of cleaned fascicles for the three pathways of interest in patients and control participants

| Subject | LGN ↔ hMT+ | | Crossing hMT+ | | SC ↔ hMT+ | |
| | Ipsi-lesional | Contra-lesional | Left → right | Right → left | Ipsi-lesional | Contra-lesional |
|---|---|---|---|---|---|---|
| Blindsight positive patients | | | | | | |
| PB1 | 75 | 115 | 9 | no | 12 | 12 |
| PB2 | 19 | 196 | 6 | no | no | 18 |
| PB3 | 50 | 67 | 24 | 24 | 17 | 17 |
| PB4 | 19 | 315 | no | no | no | 15 |
| PB5 | 93 | 83 | 7 | 19 | 14 | 14 |
| PB6 | 12 | 64 | 13 | 15 | no | 17 |
| PB7 | 397 | 17 | no | no | 16 | 8 |
| PB8 | 87 | 37 | 12 | 16 | 20 | 17 |
| PB9 | 635 | 53 | no | no | 16 | no |
| PB10 | 32 | 29 | no | no | no | no |
| PB11 | 291 | 47 | 9 | 18 | 12 | no |
| PB12 | 194 | 17 | 17 | 13 | 15 | no |
| Blindsight negative patients | | | | | | |
| PN1 | 157 | 19 | no | no | 15 | 8 |
| PN2 | 17 | 226 | 13 | 15 | 7 | 21 |
| PN3 | 351 | 89 | no | no | 19 | 16 |
| PN4 | no | 101 | no | no | no | 19 |
| PN5 | 15 | 122 | no | no | no | 13 |
| Controls | | | | | | |
| C1 | 308 | 339 | 19 | 14 | 14 | 18 |
| C2 | 619 | 269 | no | no | 39 | 17 |
| C3 | 57 | 59 | 8 | 16 | 18 | no |
| C4 | 176 | 114 | 8 | 6 | 18 | 16 |
| C5 | 84 | 30 | 5 | 8 | 17 | 16 |
| C6 | 57 | 19 | no | no | 15 | 16 |
| C7 | 78 | 46 | no | no | 9 | no |
| C8 | 498 | 182 | 19 | 14 | 35 | 14 |
| C9 | 653 | 62 | 57 | 52 | 31 | 9 |

(1) Ipsilateral pathway between LGN and hMT+ (2) Pathway between hMT+ bilaterally via the corpus callosum (3) Ipsilateral pathway between SC and hMT+. Results are shown separately for the intact and damaged 'ipsi-lesion' hemispheres (right and left for control participants). 'no' = zero fascicles survived the cleaning process.

hemisphere) and 1.6% for MD (range = $0.56 \times 10^{-3}$–$0.91 \times 10^{-3}$, Mean MD = $0.73 \times 10^{-3} \pm 0.03 \times 10^{-3}$ left, $0.72 \times 10^{-3} \pm 0.03 \times 10^{-3}$ right hemisphere).

White matter tract MD in patients was consistent with the findings for FA. In blindsight positive cases, mean MD was $0.81 \times 10^{-3} \pm 0.07 \times 10^{-3}$ in the damaged hemisphere, and $0.73 \times 10^{-3} \pm 0.05 \times 10^{-3}$ in the intact hemisphere (laterality = 9.6%). Conversely, blindsight negative patients had a mean MD of $1.05 \times 10^{-3} \pm 0.22 \times 10^{-3}$ in the damaged hemisphere, compared to $0.77 \times 10^{-3} \pm 0.05 \times 10^{-3}$ on the intact side, representing a laterality of 27.0%.

The differences between blindsight patients and lesion side can be illustrated using a two-way ANOVA of the FA values within the geniculate-hMT+ tract. While there was no significant effect of blindsight status (positive or negative; F = 2.6, p = 0.13), there was a highly significant effect of lesion

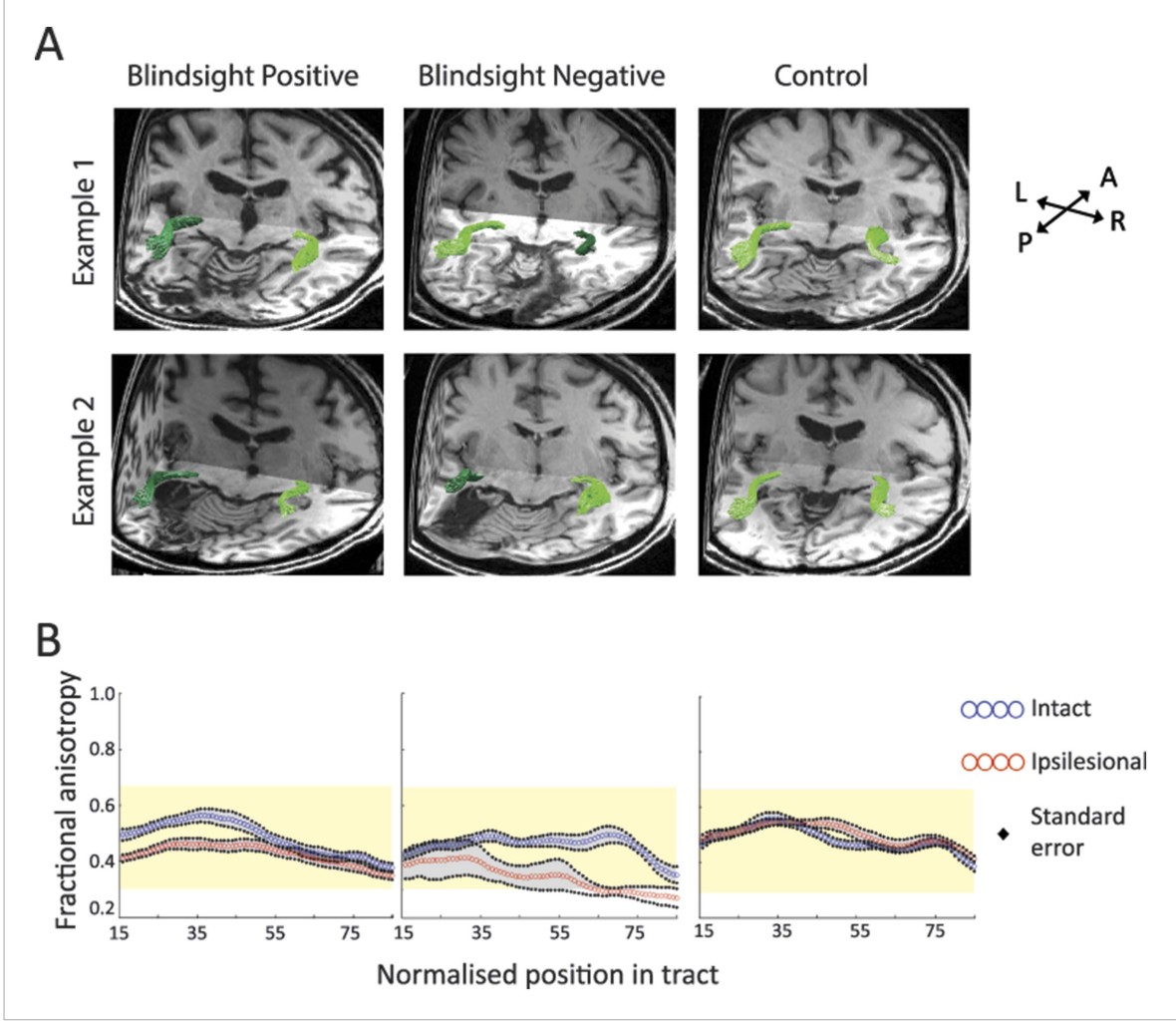

**Figure 2**. (**A**) 3-D representations of ipsilateral tracts between the LGN and hMT+. Examples are shown for blindsight positive patients PB9 and PB8, blindsight negative patients PN2 and PN3 and control participants C8 and C4. Dark green tracts are in the ipsilesional damaged hemisphere, light green tracts are in the intact hemisphere and controls. Tracts are overlaid on a 3-D representation of participant's structural T1-weighted images. (**B**) Average FA along the ipsilateral geniculate-hMT+ pathways of blindsight positive patients, blindsight negative patients, and controls. Blindsight positive patients show a slight reduction in anisotropy over the proximal half of the ipsilesional pathway, although the distal half shows no notable difference to the intact hemisphere. Blindsight negative patients show a marked reduction in FA in the damaged hemisphere beyond the 35th node, continuing to the end of the tract. Control participants show similar results for both hemispheres (right hemisphere blue, left hemisphere red), with FA close to 0.5 throughout. The control range for this pathway is displayed in yellow on all charts.

side (ipsilateral or contralateral; F = 35.7; p < 0.00005) and interaction, suggesting a differential effect of the lesion in the two patient groups (F = 5.1; p = 0.04). This effect was even stronger for MD: significant effect of blindsight status (F = 9.2; p < 0.01), significant effect of lesion side (F = 35.7; p < 0.00005) and interaction (F = 12.4; p < 0.005).

Since estimates of distinct pathways frequently overlapped (*Figure 3A*), the slight laterality in patients may, at least in part, be driven by an overlap with degenerated optic radiations supplying damaged V1. *Figure 3A* shows how this overlap can occur in the early-mid portions of the geniculate-hMT+ pathway. In the central nervous system, anterograde (Wallerian) or retrograde neuronal degeneration can occur following axonal injury. Consequently, the integrity of optic radiation fibres innervating damaged V1 would be abnormal throughout their course (similar to *Danek et al., 1990*). Where overlap with such fibres occurs, the dMRI measurements would not distinguish between separate axonal bundles due to limitations in spatial resolution (restricted here to 2 mm isotropic

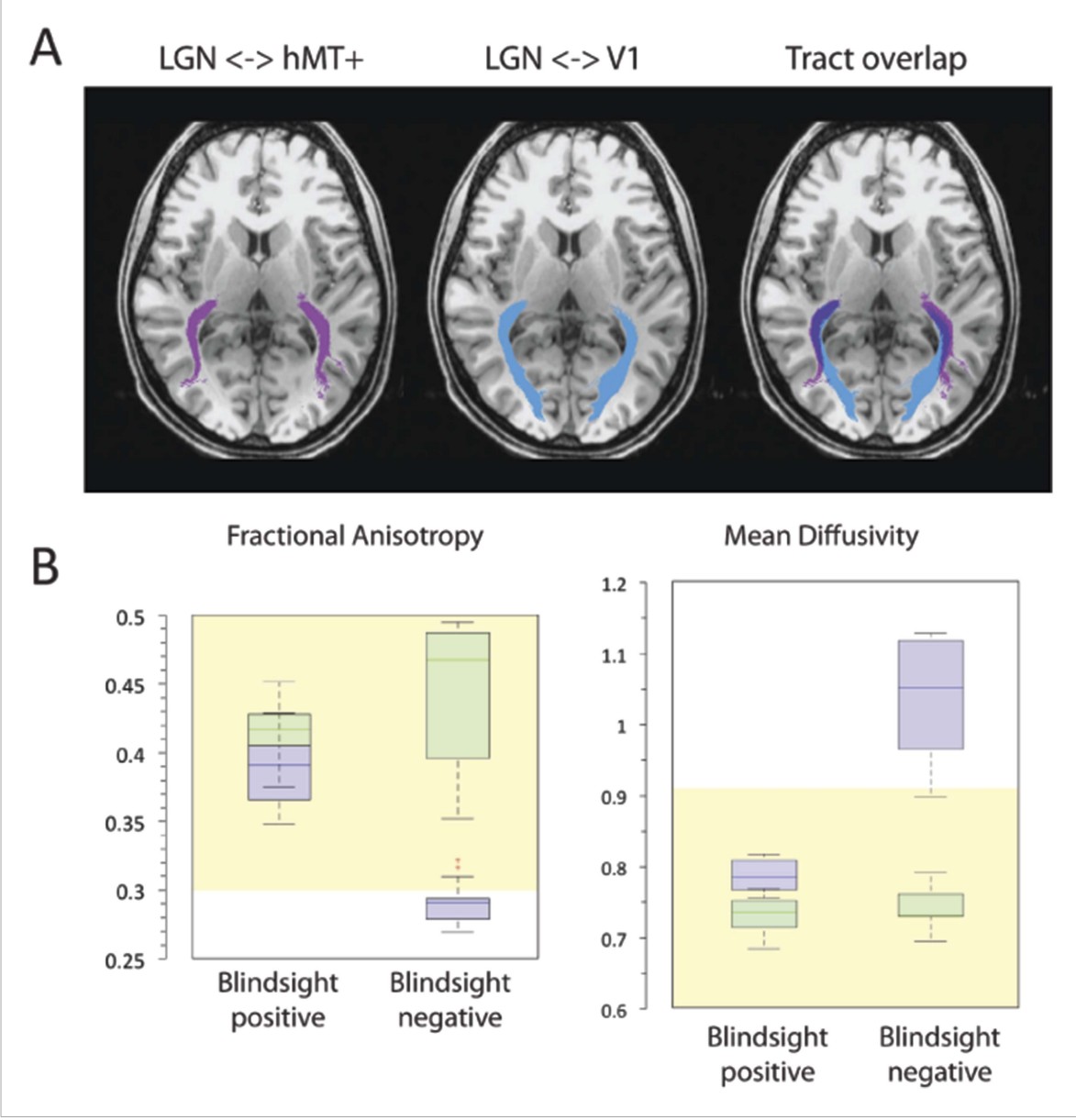

**Figure 3**. (**A**) Normal ipsilateral tracts between the LGN and hMT+, and the LGN and V1 demonstrate a proximal region of overlap. Tracts are demonstrated in a control participant, C2, comparing ipsilateral connections between the LGN and hMT+ (pink) and the LGN and V1 (blue). When these pathways are superimposed, there is a significant region of overlap in the proximal portion of these pathways. In cases of V1 damage where there is retrograde degeneration, this overlapping region of the geniculate-hMT+ pathway may become contaminated by degenerated tracts in the V1 pathway. (**B**) Box plots comparing FA and MD in the distal portion of the geniculate-hMT+ pathway, in blindsight positive and negative patients. The ipsilesional hemisphere is shown in purple, and the intact hemisphere in green. Blindsight positive patients show significant overlap in the FA of the distal portion of this pathway in the damaged and sighted hemispheres. There is a slight increase in MD in the damaged hemisphere, however this is not marked and both measures fall within the control range. In comparison, blindsight negative patients show a marked difference in FA and MD for this pathway in the damaged and sighted hemispheres. The ipsilesional measures extend beyond the control range, implying that they are pathological and significantly impaired. Adjacent values are defined as the lowest and highest observations that are still inside the region defined by the following limits: Lower Limit = Q1 − 1.5 x IQR, Upper Limit = Q3 + 1.5 x IQR. The age-matched control FA and MD range for this pathway are displayed in yellow.

voxel size). Thus measures of the geniculate-hMT+ tract could become contaminated with overlapping degenerated optic radiation fibres. It may therefore be useful to measure the diffusivity spanning only the distal portion of the geniculate-hMT+ tract, which has branched away from the large geniculate-V1 radiation bundle. This may represent a purer measure of the pathway, removing

artefacts due to overlapping tracts. If the pathway to hMT+ were actually damaged, one would still expect this measure to reflect the damage.

Figure 3B shows the microstructure of just the distal portions of the geniculate-hMT+ pathway (region between 60% and 85% of the total tract length from the LGN). Mean FA in blindsight positive patients was $0.39 \pm 0.06$ in the damaged hemisphere and $0.42 \pm 0.04$ in the intact hemisphere, corresponding to a laterality of only 7.7% (mean MD = $0.79 \times 10^{-3} \pm 0.06 \times 10^{-3}$ vs $0.73 \times 10^{-3} \pm 0.06 \times 10^{-3}$, laterality = 7.6%). Individual data also confirmed that the FA or MD standard deviation in each blindsight positive patient overlapped with the opposite hemisphere. Overall, this implies a less pronounced impairment to the white matter than suggested by the entire extent of the tract.

In blindsight negative patients, mean FA in the distal portion of this tract was $0.29 \pm 0.07$, compared to $0.44 \pm 0.05$ in the intact hemisphere (laterality = 51.7%), mean MD = $1.04 \times 10^{-3} \pm 0.23 \times 10^{-3}$ vs $0.74 \times 10^{-3} \pm 0.07 \times 10^{-3}$, (laterality = 28.8%). Therefore, unlike the blindsight positive group, blindsight negative patients still showed a relatively marked and significant drop in FA and increase in MD throughout this purer geniculate-hMT+ tract ROI when compared to the intact hemisphere. Although the effect of blindsight status on mean FA within this distal portion was not significant (F = 0.7; p = 0.42), both the effect of lesion side (F = 48.6; p < 0.00001) and the interaction (F = 15.6; p < 0.0005) were highly significant. The effect of blindsight status on mean MD was significant (F = 7.9; p = 0.01), as were the effect of lesion side (F = 31.4; p < 0.0001) and the interaction (F = 15.6; p = 0.001). This difference can also be appreciated in brain images by inspecting the tracts in the white matter, and their corresponding FA and MD maps (Figure 4). Only the blindsight negative patients (Figure 4, lower portion) possess tracts in the damaged hemisphere that appear to traverse a region of white matter displaying very abnormal FA and MD levels. Thus, although fascicles successfully propagated through this region, they passed through regions of profoundly abnormal, damaged tissue (see also Figure 4—figure supplement 1 for greater detail).

To ensure that any differences in tract microstructure between blindsight positive and negative patients were not driven by differences in grey matter volume in hMT+, the volume was directly compared both between hemispheres and groups. In the blindsight positive patients the mean grey matter volume was 159 mm$^3$ $\pm$ 59 in the ipsilesional hemisphere and 167 mm$^3$ $\pm$ 55 in the intact hemisphere. In blindsight negative patients the equivalent numbers were 135 mm$^3$ $\pm$ 71 and 169 mm$^3$ $\pm$ 83. There was no significant effect of blindsight status (F = 0.1; p = 0.8), lesion side (F = 1.7; p = 0.2) or interaction (F = 1.0; p = 0.3), indicating that differences in grey matter volume within hMT+ are unlikely to have affected the results significantly.

## Alternative pathways cannot account for the presence of blindsight

So far we have observed a difference in geniculate-hMT+ tract properties between blindsight positive and negative patients, indicating that this pathway might be a candidate for blindsight. Visual motion information could, however, travel via other pathways such as a transcallosal tract connecting left and right hMT+ (Figure 5A–C), or a pathway connecting the superior colliculus and hMT+ (Figure 5D–F). Next we tested whether these alternate tracts could account for blindsight.

Interestingly, in a number of blindsight positive patients it was not possible to identify either a pathway between hMT+ and the superior colliculus, between hMT+ in the two hemispheres, or both. Similarly, intact pathways between these regions were present in blindsight negative cases. Overall both pathways generated approximately 10-fold fewer fascicles than the geniculate-hMT+ pathway (mean 18.9/18.7 vs 202.8 in controls, and mean 15.6/15.0 vs 122.7 in blindsight positive patients). Furthermore, the collicular-hMT+ tracts appeared less consistent in shape and trajectory between individuals.

## Interhemispheric hMT+ tracts

Crossing tracts between hMT+ bilaterally were identified in only 6/12 patients with blindsight and 6/9 controls (Table 1, columns 3–4). As expected, pathways always crossed to the opposite hemisphere via the corpus callosum (Figure 5A–C). Where present, tracts also appeared to possess normal FA and MD. In blindsight positive cases, mean FA was $0.64 \pm 0.07$ (mean $\pm$ s.d.) and mean MD $0.70 \times 10^{-3} \pm 0.03 \times 10^{-3}$, averaged along the entire tract for both directions (left to right, and right to left), and across participants (Figure 6A). These values were similar to controls (Figure 6C, mean FA = $0.64 \pm 0.07$, mean MD = $0.67 \times 10^{-3} \pm 0.05 \times 10^{-3}$).

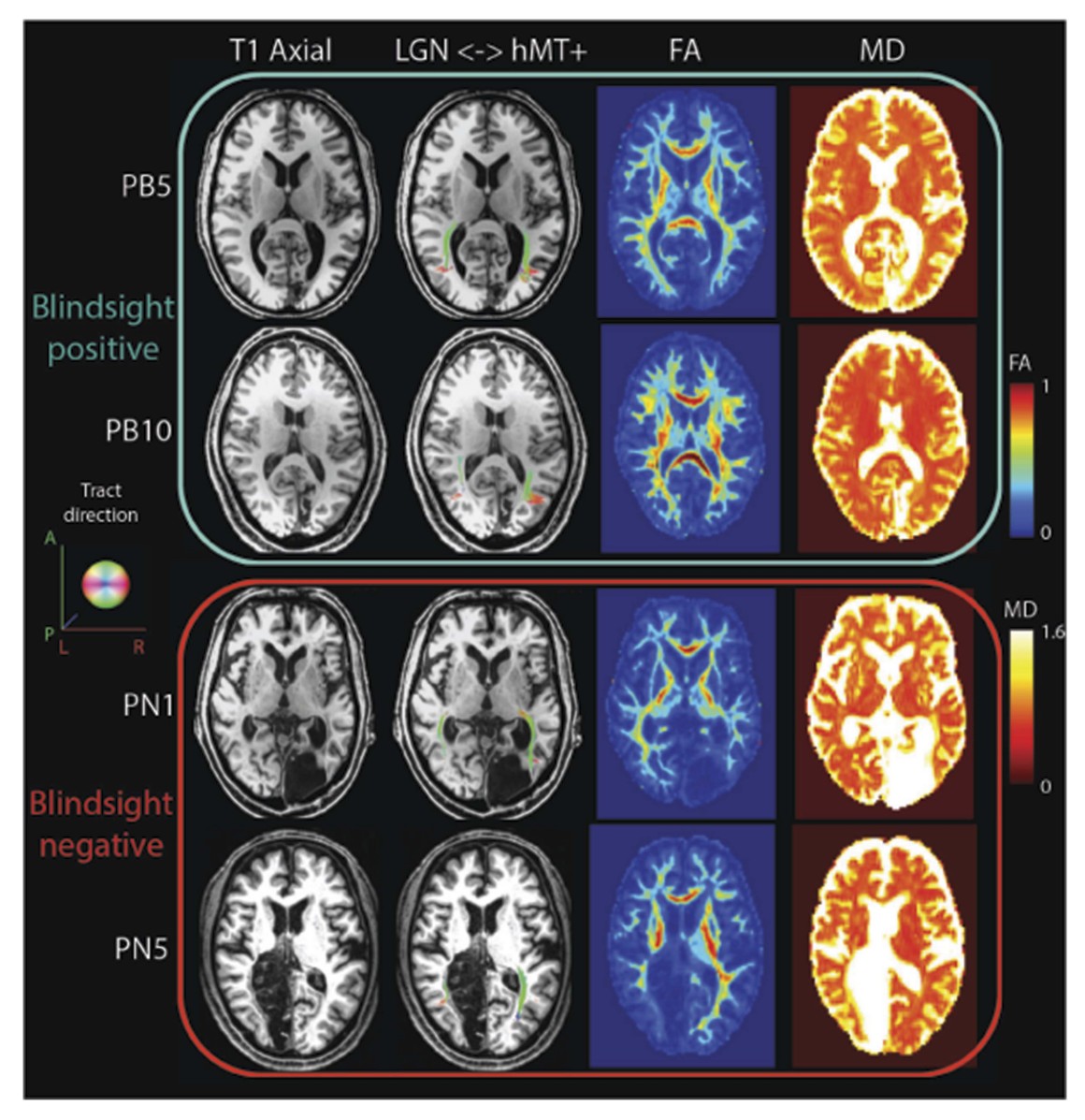

**Figure 4**. FA and MD maps in blindsight positive and negative patients, demonstrating the spatial relationship with the geniculate-hMT+ pathways. Individual results are shown for two blindsight positive patients PB5, and PB10 and two blindsight negative patients, PN1 and PN5. All four patients showed bilateral ipsilateral fascicles between the LGN and hMT+, including the damaged hemisphere (column two). In the damaged hemisphere of blindsight positive patients the region directly underlying tracts corresponds to relatively intact MD and FA measures, not notably different from the intact hemisphere. However, both blindsight negative patients have tracts in the damaged hemisphere that traverse a region of tissue with markedly abnormal FA and MD values (columns three and four).

The following figure supplement is available for figure 4:

**Figure supplement 1**. Zoomed in view demonstrating ipsilesional geniculate-hMT+ tracts with the corresponding T1-weighted structural, FA and MD maps.

Only a single blindsight negative patient showed an interhemispheric connection between hMT+ bilaterally (PN2; *Table 1*, *Figure 5B*), and in this case, the tract appeared to be largely intact and remained within the control FA range (0.28–0.91), with mean FA = 0.59 and MD = 0.73 x $10^{-3}$ (see *Figure 6B* for FA plots along this path).

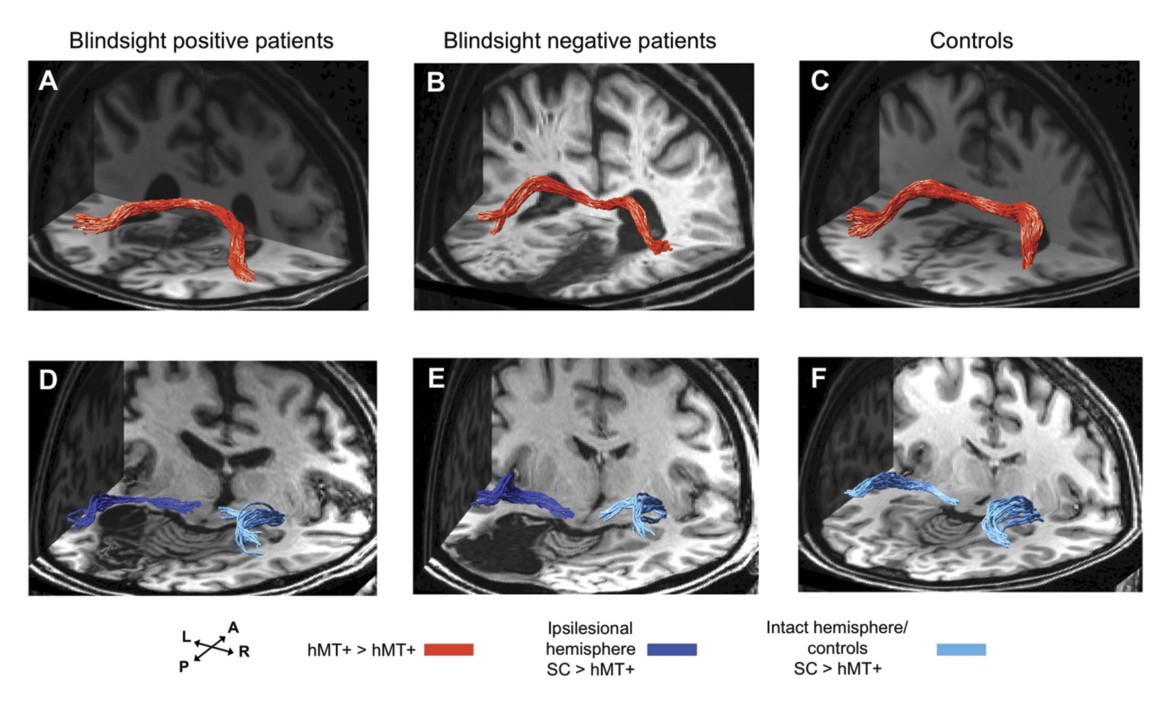

**Figure 5**. 3-D representations of interhemispheric tracts between hMT+ bilaterally and ipsilateral tracts between SC and hMT+. (**A–C**) Interhemispheric hMT+ tracts in blindsight positive patient, PB3, a blindsight negative patient, PN2 and a control participant, C9. (**D–F**) Ipsilateral collicular-hMT+ tracts in blindsight positive patient, PB8, a blindsight negative patient, PN3 and a control participant, C2. Red tracts represent crossing, interhemispheric connections between hMT+ bilaterally. Dark blue tracts are connections between SC and hMT+ in the ipsilesional damaged hemisphere, light blue tracts show the same collicular-hMT+ pathway in the intact hemisphere, and in controls. Tracts are overlaid on a 3-D representation of participant's structural T1-weighted images.

## Superior colliculus tracts

Similarly, the collicular-hMT+ pathway, could not be tracked in all patients with blindsight, and was demonstrable in some blindsight negative individuals. Specifically, these pathways were tracked in the damaged hemisphere of 8/12 blindsight positive patients (*Figure 5D*). The same proportion of patients had tracts in their intact hemisphere, although not necessarily in the same cases (*Table 1*, columns 5–6). In comparison, this pathway was present in 3/5 blindsight negative patients (*Figure 5E*). Control participants showed these pathways in all nine cases on the left, and 7/9 on the right (*Figure 5F*). Of the patients demonstrating this pathway, mean FA was $0.40 \pm 0.05$ in the damaged hemisphere of blindsight positive cases, vs $0.46 \pm 0.03$ on the intact side, with some regions of overlap along their trajectory (laterality of 13.9%, see *Figure 6D* for FA plots). Mean MD was $0.78 \times 10^{-3} \pm 0.08 \times 10^{-3}$ vs $0.69 \times 10^{-3} \pm 0.04 \times 10^{-3}$ (laterality = 10.8%).

In blindsight negative patients (*Figure 6E*), the pattern of FA was more variable along its trajectory compared to other tract profiles (i.e. *Figures 2B, and 6A–C*). Collapsed along the pathway, mean FA was $0.37 \pm 0.05$ vs $0.42 \pm 0.03$ on the intact side (laterality = 14.0%), and MD was $0.83 \times 10^{-3} \times 10^{-3} \pm 0.13$ vs $0.77 \times 10^{-3} \pm 0.07 \times 10^{-3}$ (laterality = 7.5%). In fact, this laterality and distal drop in FA was strongly influenced by data from one patient (PN1). The other two patients (PN2 and PN3) showed a similar microstructure in the distal portion of their collicular tracts in both hemispheres (t = 1.7, p = 0.2, mean FA = 0.32 vs 0.37) despite a significant laterality in their geniculate-hMT+ pathway (t = 12.2, p = 0.01, mean FA = 0.34 vs 0.48). This implies that intact tracts from the superior colliculus can occur in blindsight negative patients. However, it is worth noting that the FA in one of these two patients (PN2) does drop below the control range despite a normal laterality, reaching an FA of 0.22 between nodes 69 and 79 (control range = 0.26–0.62, see *Figure 6E*).

Statistical comparison of these values is complicated because only 5/12 of the blindsight positive and 3/5 blindsight negative patients had tracts in both hemispheres. This makes the comparison

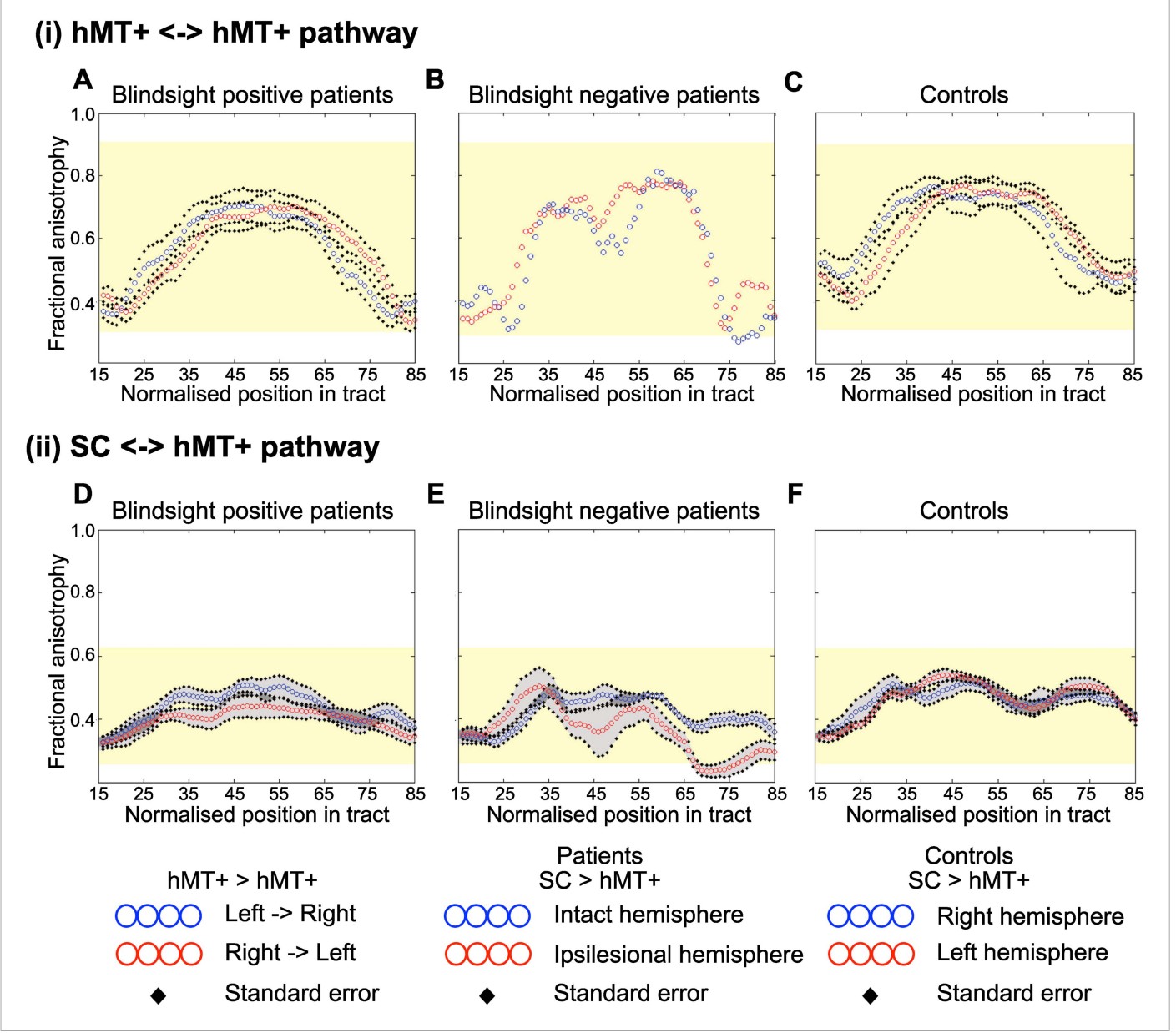

**Figure 6**. Average fractional anisotropy along the Interhemispheric hMT+ pathway and ipsilateral pathway between SC and hMT+. (**A**) Blindsight positive patients show a similar FA to controls along the length of interhemispheric hMT+ pathways. (**B**) Blindsight negative patient, PN2, also shows a similar FA to controls along the length of this pathway. (**C**) Control participants show a normal peak in FA at the centre of the interhemispheric hMT+ pathway, representing the high degree of anisotropy at the corpus callosum. (**D**) Blindsight positive patients show a similar FA in the ipsilesional collicular-hMT+ pathway as the intact hemisphere and controls. (**E**) Blindsight negative patients show a slight drop in mean FA in the distal third of the ipsilesional collicular-hMT+ pathway. (**F**) Control participants show a fairly constant FA along the length of the collicular-hMT+ pathway, around 0.4. The control range for each pathway is displayed in yellow.

between the ipsi- and contra-lesional hemispheres problematic. However, a comparison of FA and MD in just the ipsilesional hemisphere indicated no significant difference in either measure between the two groups (FA: t = 0.9; p = 0.4; d.f. = 9; MD: t = 0.8; p = 0.45; d.f. = 9).

## Relationship between blindsight performance and tract microstructure

The analyses thus far have addressed group differences by division of patients into blindsight positive and negative groups. However, even within the blindsight positive group, there is considerable

variability in performance. Therefore, the percentage of correct responses in the blindsight task was correlated with the measures of mean FA and MD extracted from the three tracts of interest. In the geniculate-hMT+ tract, 16 patients were included as one blindsight negative patient did not have an identifiable tract, and for this tract only the distal portion was used. *Figure 7A* shows the correlation for FA and MD from this distal portion of the tract across all patients, with blindsight negative indicated by the open symbols and blindsight positive indicated by the filled symbols. Although both correlations were in the predicted direction, positive for FA and negative for MD, neither was significant (r = 0.44; p = 0.09 for FA; r = −0.48; p = 0.06 for MD). Since age can be a confounding factor in tract microstructural properties, the partial correlation coefficients, accounting for age were also calculated, but did not differ from the full correlations. Neither the collicular-hMT+ (r = −0.03; p = 0.93 for FA; r = −0.10; p = 0.77 for MD) nor the interhemispheric hMT+ (r = −0.33; p = 0.39 for FA; r = 0.26; p = 0.50 for MD) tracts showed any correlation with behaviour.

## Lesion size and location

In addition to performing tractography between pre-defined regions of interest, a useful and unbiased approach to understand why certain patients have blindsight and others do not is to quantify lesion extent and location from the T1-weighted anatomical image. This is particularly valuable in larger patient cohorts given the heterogeneity of damage in such groups. *Figure 8* shows the total lesion volume and distribution of damage across all patients. Average lesion volume in the blindsight positive group was 13,461 mm$^3$ ± 7101 mm$^3$ s.d., compared to 36,923 mm$^3$ ± 23,035 s.d. in the blindsight negative group. On average, blindsight negative patients had lesions approximately 2.5 times larger than the blindsight positive group, although *Figure 8B* shows the overlap of occipital lobe damage between patients. There was a significant association between the extent of occipital lobe damage and the microstructural measures of ipsilesional geniculate-hMT+ pathways across all patients (FA: r = −0.59, p = 0.015; MD: r = 0.63, p = 0.01). This reinforces the likelihood that reduced FA and increased MD in blindsight negative individuals reflects an involvement of surrounding white matter pathways in occipital lesions.

There was no clear association between blindsight function and the presence of additional, non-occipital damage. As seen in *Figure 8B*, the damage in some patients extended to other regions, including the temporal (PB8, PN3, and PN5) or parietal lobe (PN4 and PN5). However, the pattern of such damage was not associated with a particular group. Furthermore, only one participant showed

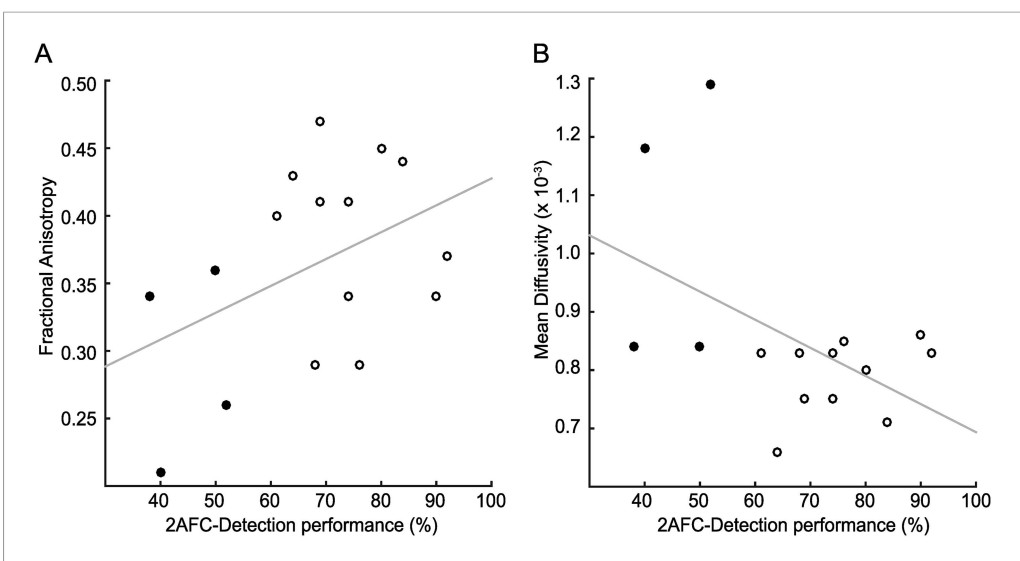

**Figure 7**. Correlation of tract microstructure in the distal region of the geniculate-hMT+ pathway with behavioural performance on the contrast detection task. In both plots the filled symbols represent the values for the blindsight negative patients while the open symbols are from the blindsight positive patients. (**A**) shows the data for the FA values (r = −0.43; p = 0.09) and (**B**) shows the corresponding values for MD (r = −0.48; p = 0.06).

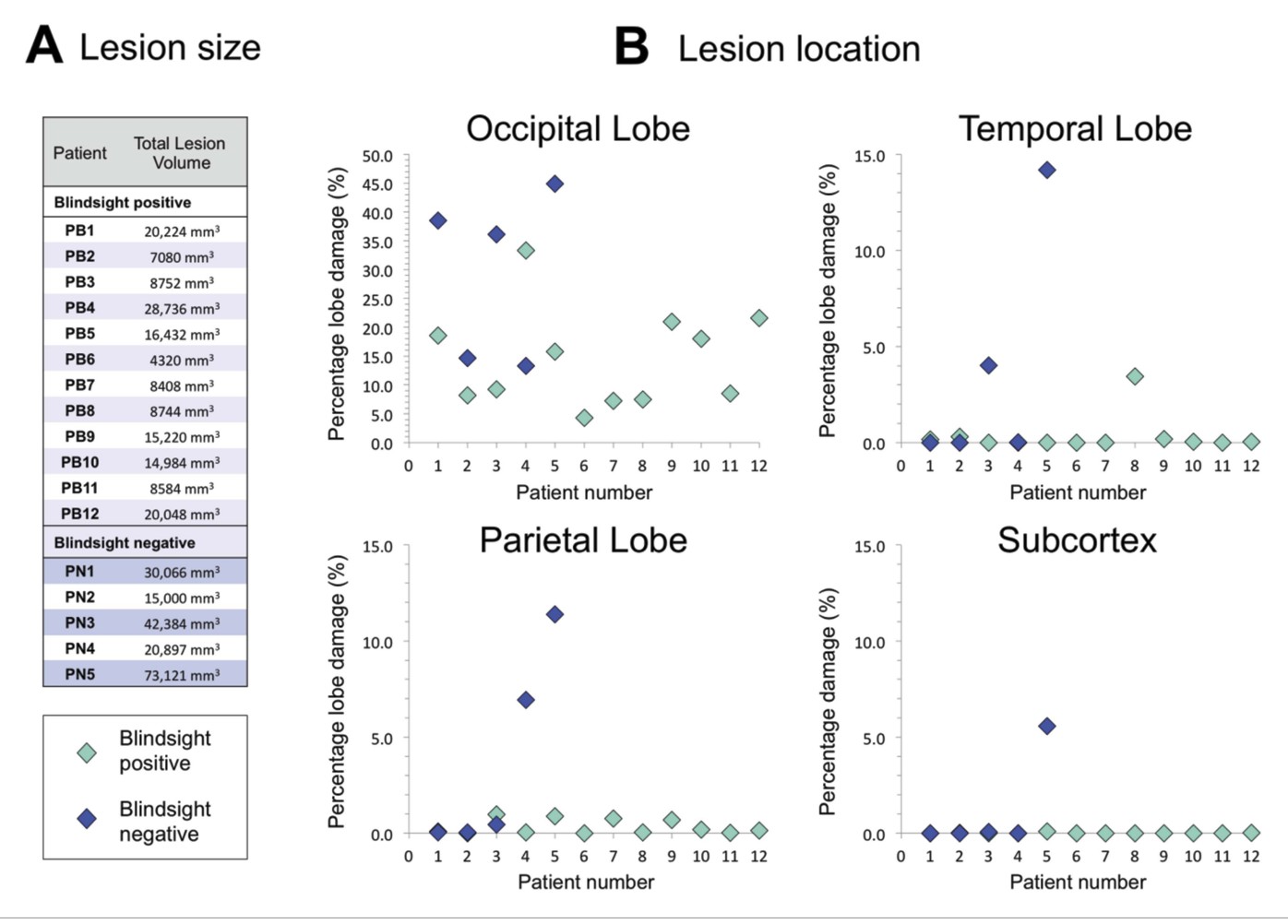

**Figure 8**. Comparison of lesion size and location in blindsight positive and negative patients. (**A**) Lesion size is given for each patient, and demonstrates a wide range of volumes in both patient groups. (**B**) Lesion location shows the proportion of lobe damage in each patient, within the occipital, temporal, and parietal lobes, as well as the subcortex. Subcortex incorporates the thalamus (including LGN and pulvinar), striatum, and superior colliculi, with an approximate unilateral volume of 50,000 mm³. Only one patient, PN5, demonstrated some involvement of this region, including the ipsilesional LGN and pulvinar, but not the superior colliculi.

evidence of significant subcortical pathology (PN5), seen to extend to a region including the ipsilesional LGN and pulvinar, although the superior colliculi appeared intact.

At least three patients with blindsight showed complete destruction of calcarine cortex or its underlying white matter (PB1, PB4, PB10). Similarly there were blindsight negative cases with small regions of V1 apparently intact (PN2, PN4). The majority of cases with lesions affecting less than 20% of the occipital lobe had some small area of V1 sparing, which usually corresponded to the occipital pole, or the anterior tip of the calcarine sulcus.

## Discussion

This is the first study to perform dMRI-based tractography and a comparison of the microstructural tissue properties of visual pathways in a group of patients with V1 damage, categorised according to blindsight function. All patients were labelled as blindsight positive or negative according to their ability to detect a highly salient stimulus in their blind hemifield. By combining the results from these psychophysical and MRI techniques, it has been possible to directly relate residual visual function to the underlying properties of the visual pathways.

## A direct geniculate pathway consistently supporting blindsight function

The principal finding was that all patients with blindsight function showed intact, undamaged tracts between the LGN and hMT+ in the hemisphere with V1 damage. This is consistent with our recent fMRI report of motion processing after V1 damage (*Ajina et al., 2015a*). A similar direct geniculate pathway was identified in all age-matched controls, and is consistent with neuroanatomical investigation in the macaque (*Sincich et al., 2004*). This was not the case in blindsight negative patients, where such geniculo-extrastriate tracts were either absent or demonstrated significant impairment in mean diffusivity and fractional anisotropy. This has two important implications that support geniculate-extrastriate connections in blindsight. (1) It is possible that intact connections between the LGN and hMT+ are *sufficient* for blindsight, since no patients with blindsight function demonstrated an absence or impairment in these pathways. (2) Intact connections between the LGN and hMT+ may also be *necessary* for blindsight function. This is supported by the results in blindsight negative patients, as none of the patients without blindsight function possessed normal, intact connections in this pathway. However, since the current study only investigated a limited number of potential pathways, it is possible that other, unexplored, pathways could also underlie blindsight function in these patients.

## Intact collicular or interhemispheric pathways are unlikely to underlie blindsight

Unlike direct geniculate connections, there were examples of patients with blindsight function who had absent or impaired collicular and interhemispheric pathways. Similarly, there were blindsight negative cases with apparently undamaged connections between these regions. These results suggest that neither of these other putative blindsight pathways have a major role in blindsight function, but support the argument that in blindsight positive cases, another pathway must have facilitated visual performance. However, these pathways may still contribute to blindsight in some circumstances.

## Relationship between blindsight performance and tract microstructure

Only the geniculate-hMT+ tract showed a marginal correlation of behavioural performance with MD and FA that was in the correct direction: improved performance correlated positively with FA and negatively with MD. However, neither of these correlations was significant. Although the current study provides the largest participant group reported to date, the considerable variability in values for tract microstructure means that there may not be sufficient power to find differences, particularly for smaller tracts.

## Pulvinar pathways to hMT+

Here we have selected three pathways with very distinct trajectories to investigate. There are, however, other potential pathways by which blindsight information could be processed, the most prominent of which is the one from the medial portion of the inferior pulvinar to MT identified in multiple primate species (*Maunsell and van Essen, 1983*; *Warner et al., 2012*). There is some debate as to the relative strength of the connections to hMT+ from LGN or the inferior pulvinar (*Sincich et al., 2004*; *Warner et al., 2010*), although there is evidence that the pulvinar connection is stronger during early development (*Warner et al., 2015*).

There are several practical reasons why the pulvinar connection with hMT+ has not been quantified in the current study. Firstly, the most commonly described pathway is a di-synaptic pathway from colliculus to hMT+ via the inferior pulvinar (*Lyon et al., 2010*). Thus, this corresponds to the collicular-hMT+ pathway examined here that was both difficult to track, but also showed reduced microstructure in a number of blindsight positive patients. A direct, retinorecipient pathway from the inferior pulvinar has been described in the marmoset (*Warner et al., 2010*), suggesting that it would also be worth considering only a connection between inferior pulvinar and hMT+. A recent human tractography study considered tracts from both LGN and pulvinar to hMT+ as part of an investigation into the visual pathways in amblyopia (*Allen et al., 2015*). Quantification of the tract microstructure found that they were very similar with almost identical values for both FA and MD. Thus, at the current resolution of 2 mm isotropic voxels, dissecting apart these two tracts may be impossible, due to the proximity of the thalamic structures. In future studies, higher spatial resolution

may help to disentangle these two important pathways. Thus, we cannot completely rule out the presence of an intact direct pathway from the retina to hMT+ via the pulvinar.

## Important differences compared to existing tractography studies

All three of the pathways studied here have been previously investigated in case studies, although they have never been compared in the same patients. Two studies investigated the pathways underlying motion (*Bridge et al., 2008*) or affective blindsight (*Tamietto et al., 2012*) in blindsight patient GY. The motion study reported a direct ipsilateral connection between LGN and hMT+ in the damaged hemisphere, similar to the results for blindsight positive patients here. However, GY also showed unusual patterns of connectivity that may be indicative of plasticity. These included a cortico-cortical callosal connection between hMT+ bilaterally (tested here, but absent in 6/12 blindsight positive patients) and a crossing pathway between LGN in the undamaged hemisphere and ipsilesional hMT+. In both cases these unusual pathways were largely demonstrable in controls, although GY showed a considerably greater number of fascicles (*Bridge et al., 2008*).

The only study to investigate collicular pathways was in patients following hemispherectomy, two of whom had attentional blindsight (*Leh et al., 2006*). Only patients with blindsight showed crossing tracts between the superior colliculus in the damaged hemisphere and regions of the intact hemisphere, as well as strong ipsilateral connections in the damaged hemisphere. These crossing tracts were seen in some control participants, although were arguably less prominent and were therefore also taken as a possible indicator of plasticity.

The current study found no evidence to support such plasticity in adult-onset V1 damage and blindsight. Furthermore, additional evidence against a necessary transcallosal connection comes from cases of bilateral cortical damage with significant fMRI hMT+ activity and blindsight (*Bridge et al., 2010*). Where occipital damage is bilateral, the corpus callosum undergoes profound degeneration and is unlikely to provide useful visual information (*de Gelder et al., 2008*).

One possible explanation for some of these differences is the age of brain injury onset, since damage acquired in childhood may lead to greater plastic changes (*Anderson et al., 2011*; *Tinelli et al., 2013*). GY sustained his brain injury aged 8 years, and the hemispherectomy patients sustained severe structural brain damage at birth or in early childhood, despite undergoing resective surgery later in life. Both studies identified increased interhemispheric connectivity in blindsight, unlike the cases of cortical blindness and patients in the current study, all of whom sustained damage in adulthood. This could be consistent with an increased propensity for plasticity in the corpus callosum, which continues to grow in cross-sectional area until early adulthood (*Keshavan et al., 2002*).

The other factor to consider is how blindsight is assessed, and the *type* of blindsight present. It has been argued in the past that different forms of blindsight may be mediated by distinct anatomical pathways or structures (*Danckert and Rossetti, 2005*). For example, collicular processing may be involved in 'action' or 'attention' blindsight, whilst the LGN is implicated in perceptual characteristics, described as 'agnosopsia' (*Zeki and Ffytche, 1998*). The definition of blindsight differs considerably between tractography studies, ranging from comparable 2-AFC testing (*Bridge et al., 2010*) to navigational tests (*de Gelder et al., 2008*) and indirect or 'attentional' blindsight (*Leh et al., 2006*). In particular, patients with extensive cortical damage beyond V1 appear to lack any awareness or direct response to blind field stimulation (*Tomaiuolo et al., 1997*; *Leh et al., 2006*; *de Gelder et al., 2008*). Indirect blindsight assessments may be more sensitive than the 2-AFC tests used here, and may rely on different structures. The only way to tackle this would be to improve consistency amongst experiments, and to include multiple methods of assessing blindsight in future work.

## Limitations of fascicle number as a useful measure in clinical populations

A significant concern highlighted from the current study is that it was possible to track robust fascicles in patients traversing regions of extremely impaired FA and MD. Indeed, it may even be the case that fascicles are biased towards narrow regions of white matter running alongside a lesion boundary. Patient PN1, for example, showed almost ten times more fascicles in the geniculate-hMT+ pathway of his damaged hemisphere compared to his intact side, even though tracts quite clearly passed through a region of abnormal (damaged) tissue (*Figure 4*). These tracts are unlikely to be functional, as indicated by the negative psychophysical performance.

Emphasis on fascicle numbers without considering the underlying microstructure and pathway viability is therefore problematic. Indeed, there are many reasons why fascicle numbers provide unreliable measures of true axonal projections and function (*Jones et al., 2013*; *Pestilli et al., 2014*). Even if the 'true' fibre count is uniform, the number of reconstructed fascicles may differ due to the length, curvature, and degree of branching present (*Jones and Cercignani, 2010*). Such variability was apparent here, as even control participants showed notable differences in fascicle numbers between hemispheres and individuals. Two of the key early papers on blindsight have focused on this measure (*Leh et al., 2006*; *Bridge et al., 2008*), interpreting a quantitative difference in fascicles as suggestive of plasticity. Whilst this may be correct, any tractography algorithm with a bias for peri-lesional pathways could contribute to such findings.

### Absent fascicles do not necessarily mean an absent pathway

One of the more controversial uses of MRI diffusion tractography is to comment on the existence or absence of a specific pathway, with false positive connections particularly problematic (*Sherbondy et al., 2008*; *Gao et al., 2013*). This is not surprising if one considers that the success or failure of fascicle propagation in tractography algorithms is subject to the same limitations as the fascicle count.

In the current study, an important source of variation was the process of 'cleaning' to isolate robust and consistent tracts. If a less stringent cut-off had been used, interhemispheric hMT+ connections would be identified in 100% of controls, thus necessitating care in their interpretation. Although the interhemispheric and geniculate pathways in patients would remain unaffected, a less stringent cut-off would suggest collicular tracts were present in all blindsight positive patients. However, when visualized, these pathways containing fewer than 5 fascicles appear largely implausible, reinforcing the need for a cleaning process to improve data reliability and reduce false positives. A novel mechanism to address this in the future may be to estimate the accuracy of an estimated connectome and tract, such as using Linear Fascicle Evaluation (*Pestilli et al., 2014*).

### Conclusions

In summary this work provides strong evidence to support a direct geniculate connection to extrastriate cortex as being important for blindsight function in adult-onset V1 damage. Although alternate interhemispheric and collicular pathways were also demonstrable in a number of patients, these connections were unable to account for all blindsight cases and were often found to be intact in patients with absent blindsight performance. The results also highlight the importance of considering white matter microstructure when performing tractography in patients, which is applicable to anyone working with clinical diffusion data. Finally, appreciation of the important tracts may help to direct attempts to boost residual function through rehabilitative strategies in hemianopia.

## Materials and methods

### Participants

Seventeen patients (five female) took part in this study, of which 15 had sustained posterior circulation stroke and two had undergone benign tumour resection, see *Supplementary file 1* for details. All patients had sustained unilateral damage to V1, causing homonymous visual field loss recorded by Humphrey perimetry. Average age at the time of participation was 54.9years ± 14.4, average time after pathology onset 45 months (range 6–252 months). Nine healthy participants (54.9 ± 11.7 years old, three female) served as controls. Written consent was obtained from all participants. Control participants and patients were matched by age and sex at the time of testing. Controls all had normal or corrected-to-normal visual acuity and no history of neurological disease. Ethical approval was provided by the Oxfordshire Research Ethics Committee (Ref B 08/H0605/156). Testing was performed at the John Radcliffe Hospital, Oxford.

### Psychophysics

Psychophysical testing was conducted outside the MRI scanner, with a 60Hz CRT monitor at a distance of 68 cm. Visual stimuli consisted of a drifting achromatic Gabor patch of 5° or 8° diameter, displayed on a uniform grey background; temporal frequency 10Hz, spatial frequency 1.3 cycles/°. Five contrast levels were used: 1%, 5%, 10%, 50%, and 100%, with stimulus location restricted to the scotoma and

its corresponding location in the sighted hemifield in patients, a minimum of 3° from fixation (see *Figure 1A* for schematic representation of stimulus location).

Participants were asked to indicate whether a stimulus appeared in the first or second time-interval (*Figure 1B*). If they saw nothing, they were instructed to guess. Onset of each interval was indicated by a 500ms auditory tone, 300Hz marking onset of the first interval, and 1200Hz for the second. Visual stimuli appeared for 500 ms with jittered onset while the participant fixated on a central black cross. Stimulus contrast was altered parametrically between the five levels at random, with 20 trials per condition. The allocated interval (first or second) was also generated at random. Participants additionally performed a run of control testing, with stimuli presented to the equivalent location in their sighted visual field. Fixation was recorded throughout with an Eyelink 1000 eye tracker (SR Research Limited, Ontario, Canada), and any trials in which eye position exceeded 1° from fixation were excluded from analysis. Participants were reminded to maintain fixation, with the investigator observing this in real-time. Anyone making even a small eye movement into their damaged hemifield was given specific instruction not to do so, and it was explained that these data would have to be discarded.

The presence or absence of blindsight, or residual visual function was determined for each patient. This was defined as achieving either an average score, or a score for stimuli of 100% contrast that was significantly above chance, using a statistical threshold of p < 0.01 and a cumulative binomial distribution. This criterion led to the allocation of 12 patients as 'blindsight positive' (PB1-PB12) and five as 'blindsight negative' (PN1-PN5), see *Figure 1C* and *Table 1* for details. Classification of patients into these two groups ('blindsight positive' and 'blindsight negative') was therefore further validated using cross-validation with two other cross-validation strategies:

1. K-nearest neighbours: in each iteration one of the participants was held out and was blindly labelled (as 'blindsight negative' or 'blindsight positive') according to the label previously assigned to the majority of their k-nearest neighbours (using only performance at 100% contrast). Neighbourhood distance between the currently labelled individual and other individuals was measured in terms of their performance in all contrast levels (k was set to 5, but other values of k were also tested and results were found to be robust to choice of k).
2. A Gaussian mixture model was fit to behavioural performance data across all contrast level. Fitting was 'blind'. That is, no class labels ('blindsight positive' or 'blindsight negative') were used in fitting the multi-dimensional Gaussian distributions. Each participant was then classified into one of two groups according to their distance from the centroids of the two Gaussian distributions.

Both algorithms were implemented in scikit-learn (*Pedregosa et al., 2011*). Accuracy of the classification was evaluated relative to the labels ('blindsight positive' or 'blindsight negative') derived from the classification based only on performance at 100% contrast (also used in *Ajina et al., 2015b*).

Behavioural testing of control participants and the sighted hemisphere of patients was not possible, since the contrast task is too easy, resulting in 100% detection of even the 1% contrast stimulus.

## MRI acquisition and pre-processing

### Anatomical acquisition
A structural scan was acquired for each participant. This was a high-resolution (1 mm × 1 mm × 1 mm voxels) whole head T1-weighted MPRAGE anatomical image (TE = 4.68ms, TR = 2040ms, field of view = 200 mm, flip angle = 8°).

### Diffusion data
Diffusion-weighted data were acquired using echo planar imaging (EPI; TR = 8900 ms, TE = 91.2 ms, and voxel size of $2 \times 2 \times 2$ mm$^3$). The diffusion weighting was isotropically distributed along the 60 directions (b-value = 1500 s/mm$^2$), and a non-DWI (B0) image was acquired every 16 vol (total of four B0 volumes per image set). EPI acquisitions are prone to geometric distortions that can lead to errors in tractography. To minimise this, two image sets were acquired with the phase-encoded direction reversed, 'blip-up' and 'blip-down' (*Chang and Fitzpatrick, 1992*). This results in images with geometric distortions of equal magnitude but in the opposite direction allowing for the calculation of a corrected image (*Andersson et al., 2003*). Before correcting for geometric distortions, each image set, blip-up and blip-down, was corrected for motion and eddy-current related distortions. These corrections were performed using tools from FSL (FMRIB Centre Software Library, Oxford University;

http://www.fmrib.ox.ac.uk/fsl/), with other steps in DTI processing and tractography using the VISTALab (Stanford Vision and Imaging Science and Technology) diffusion MRI software suite. VISTALab image processing software is available as part of the open-source mrDiffusion package available at https://github.com/vistalab/vistasoft/.

The corrected 4-D NifTI DTI images from both AP (blip-up) and PA (blip-down) image sets were concatenated in time and aligned to the motion-corrected mean of the non-diffusion weighted (b = 0) images using a rigid body algorithm. dMRI images were then aligned to the T1 structural scan, which had been resampled to AC-PC orientation using an automated script.

## Diffusion MRI analysis

### Regions of interest

hMT+ masks were derived from anatomically defined probabilistic maps (Juelich atlas implemented in FSL, (*Malikovic et al., 2007*), non-linearly transformed from MNI to diffusion space for patients and controls to ensure consistency between participant groups. Average hMT+ ROI volume was $366 \pm 60$ voxels in patients, $415 \pm 60$ voxels in controls. For the LGN and superior colliculus, binary masks were created by manual inspection and drawing over the anatomical T1-weighted images (*Horton et al., 1990*), using a radiological brain atlas to aid identification of landmarks. The average LGN volume in patients measured 245 mm³ in the right, and 244 mm³ in the left. In controls, average LGN volume was 245 mm³ in the right and 236 mm³ in the left. These volumes are similar to previous reports using T1 anatomical and functional MRI scans in living humans (244 mm³ in the right, 234 mm³ in the left; (*Kastner et al., 2004*). In post-mortem human tissue, investigation has shown LGN volume ranges from 91 to 157 mm³ (*Andrews et al., 1997*). However it has been suggested that this difference may, at least in part, arise due to tissue shrinkage during post-mortem processing (e.g., *Annese et al., 2014*). Superior colliculus masks had an average volume of 203 mm³ in the right and 216 mm³ in the left of patients. In controls, superior colliculus masks were 214 mm³ in the right and 218 mm³ in the left. These are similar in size to previous studies using T1 anatomical and functional MRI scans (*Anderson and Rees, 2011*). There were no significant differences between subject groups when comparing the volume of hMT+, LGN or superior colliculus masks (LGN: F = 0.96, p = 0.4, hMT+: F = 2.0, p = 0.1, SC: F = 0.12, p = 0.9).

### Fascicle tracking

The tracking algorithm was restricted to the white matter, defined as all voxels with a FA value greater than 0.15. This method of segmentation generated a white matter mask that excluded the ventricles. This was manually inspected and edited for each participant, to ensure optimal segmentation and to remove any satellite voxels.

The diffusion tensor model is prone to error in assigning the orientation of tracking in regions where multiple populations of nerve fibres cross. Models that account for the diffusion signal as a combination of signals from different bundles of nerve fibres provide better estimates of tracking directions in these locations (*Frank, 2001*, *2002*; *Rokem et al., 2015*). Therefore, so-called fibre orientation distribution functions (fODF) were estimated in each voxel in the white matter using constrained spherical deconvolution (CSD; *Tournier et al., 2007*). A response function, representing the signal of a single coherent bundle of nerve fibres, was estimated as a lower-order ($L_{max}$ = 4) CSD fit to the signal from voxels in which FA was larger than 0.7. CSD was then fit to the entire white matter with this response function and maximum harmonic order ($L_{max}$) was set to 8. The $L_{max}$ determines the maximal order of the spherical harmonics basis set used to estimate the fODF in each voxel by the CSD model. The number of coefficients for CSD grows with $L_{max}$, as $\frac{1}{2}(L_{max}+1)(L_{max}+2)$. The $L_{max}$ was set to 8 because this number requires a number of coefficients (45) lower than the number of diffusion directions used (60) and because it has been previously demonstrated that CSD-based probabilistic tractography using $L_{max}$ = 8 generates accurate connectomes (*Yeatman et al., 2014*).

Fascicle tracking was performed on the fODFs estimated with CSD, using a probabilistic 'region to region' algorithm implemented in MRtrix (*Tournier et al., 2012*). The methodology has previously been shown to provide superior delineations of a number of known white matter tracts, in a manner that is robust to crossing fibre effects (*Tournier et al., 2012*; *Pestilli et al., 2014*). Fascicles were run from 10,000 seeds inside a union mask created by the combination of two ROIs. Tracts had to touch both ROIs and travel only within white matter to be included in the output. A curvature radius

threshold of 1 mm and step size of 0.2 mm was used. The total number of fascicles generated was constrained to a maximum of 1,000,000.

## Anatomically-informed identification of the tracts of interest

After fascicles were created for each pathway of interest, we used an anatomically informed approach to identify core-fascicles to compare across individuals (*Yeatman et al., 2012*; *Pestilli et al., 2014*; *Allen et al., 2015*). Outlier fascicles were removed from tracts in each brain to retain a core fascicle bundle representing the most conservative estimate of the tract. To identify outlier fascicles, we calculated the Mahalanobis distance of nodes in each fascicle from the core fascicle bundle. This procedure assigned a weight to each fascicle depending on its distance from the core fascicle in standard deviations of the multivariate normal distribution. If the nodes in a fascicle were more than a predetermined number of standard deviations away from the core fascicles, then the fascicle was rejected as an outlier. This was performed using an iterative process to remove fascicles located more than 2.6 standard deviations away from the core of the tract, and more than 2.8 standard deviations longer than the mean tract length, using a Gaussian distribution to represent fascicle distance and length. Where this was not possible because of a small number of sparse fascicles <10, this was interpreted as a failure to accurately track between the two regions of interest. All subsequent measures of tract integrity were then carried out using these 'cleaned' fascicle bundles. Tracts were processed using software routines part of MBA (Matlab Brain Anatomy: https://github.com/francopestilli/mba) and LiFE (Linear Fascicle Evaluation: https://francopestilli.github.io/life; (*Pestilli et al., 2014*)

## Tracts of interest

Three tracts of interest were identified in this study, all of which pass through hMT+ and have been implicated in blindsight function. Two of these pathways projected between the LGN or superior colliculus and hMT+ in the same hemisphere. The other pathway was a crossing, interhemispheric connection between hMT+ bilaterally.

## The tensor model

Although the diffusion tensor model (*Basser et al., 1994*; *Pierpaoli and Basser, 1996*) can be inappropriate for tracking, it is an accurate representation of the signal and its statistics (*Rokem et al., 2015*). This model was fitted at each voxel to derive FA and MD maps, from which the mean and variation along any fascicle bundle could be calculated. FA provides a measure of the directionality of water molecule movement, which relates to the geometric organization of axons and fascicles in each voxel (e.g., crossing, merging or 'kissing' fibres), the degree of myelination of axons in the white matter (*Beaulieu and Allen, 1994*), and their packing density (*Sen and Basser, 2005*). In cases of brain damage, a decrease in FA can be indicative of loss of structural integrity of fibre bundles (*Jones et al., 2013*), such as Wallerian degeneration (*Beaulieu et al., 1996*). Similarly, increases in MD can indicate tissue damage, for example after a cerebral infarct (*Werring et al., 2000*), and this measure is also sensitive to axon packing density and myelination (*Sen and Basser, 2005*). Since both of these measures are sensitive to a number of tissue properties, but may not specifically be attributable to any one of them (see also *Johansen-Berg, 2010* for review) we acknowledged that the precise interpretation of MD and FA is unknown. Therefore, the broad term 'white matter microstructure' is used to describe these measures.

## Tract-based statistics

In order to compare values across participants, a standardised measure was derived for each tract. The voxel-wise tensor parameters (FA and MD) were combined with the spatial information of the trajectory of tracts within the white matter to compute a tract profile. Tract profiles represented the average FA or MD of the voxels touched by the tract, weighted by the distance from the mean of the tract at each location. This was done by resampling each tract to 100 nodes, distributed equally along the length of the tract (*Yeatman et al., 2012*). The region between nodes 15 and 85 was then used to represent 'whole tract' profiles, with the proximal and distal 15 nodes ignored to remove potential contamination with grey matter voxels or partial volume effects. This clipped tract profile was used to generate all subsequent measures of mean tract FA and MD. These measures were also used to calculate 'laterality', representing the relative difference in FA or MD measures for the same tracts in opposite hemispheres.

$$\text{Laterality in patients}(\%) = \frac{\left| \text{FA}/\text{MD}_{\text{(intact)}} - \text{FA}/\text{MD}_{\text{(ipsilesional)}} \right|}{\text{FA}/\text{MD}_{\text{(ipsilesional)}}},$$

$$\text{Laterality in controls}(\%) = \frac{\left| \text{FA}/\text{MD}_{\text{(left)}} - \text{FA}/\text{MD}_{\text{(right)}} \right|}{\text{FA}/\text{MD}_{\text{(right)}}}.$$

This technique of standardisation may be preferable to the alternative method of voxel-based analysis, including Tract-Based Spatial Statistics (TBSS, *Smith et al., 2006*), which computes summary statistics on coregistered voxel skeletons. This is because individual brains show substantial variation in tract location, size, and shape, which may not be sufficiently dealt with by standard techniques that warp FA data onto a template image. This can be particularly problematic for more peripheral, long-range tracts such as those being investigated here (*Edden and Jones, 2011*).

### Statistical testing of pathway microstructure

In order to quantify differences in the microstructure of healthy controls, blindsight positive and blindsight negative patients, a number of different statistical approaches were taken, implemented in either Excel or Matlab. Firstly, a two-way ANOVA was used to investigate the effect of blindsight status (positive or negative) and side of the brain (intact or lesioned). The presence of a significant interaction was used to determine a difference in the effect of the lesion between the two groups.

Where there were not sufficient samples to compute the ANOVA, an independent samples two-way t-test was employed to quantify the effect of blindsight status (positive or negative).

### Lesion estimation

Lesion size in patients was estimated by creating lesion masks from their T1 structural scans. This required a combination of thresholding raw T1 values to isolate damaged tissue (on T1-weighted MRI scans, ischaemic pathology shows low T1 intensity) and manually drawing over unequivocal regions of damage. The 3-D lesion masks were binarised, and the total volume measured in mm³. We were also interested in estimating the distribution and extent of damage across the brain. Lobar masks were created using the MNI structural atlas in standard space for all four lobes (frontal, parietal, temporal, occipital) in both hemispheres and separately for the subcortex. Masks were transformed into individual structural space using non-linear transformation, similar to the technique to create ROIs. A region of overlap between the lesion and lobe masks was then quantified as a percentage of the total lobe volume.

## Acknowledgements

This work was supported by the Wellcome Trust (SA), the Royal Society (HB) and the NIHR Oxford Biomedical Research Centre (SA & CK). Franco Pestilli was supported by the US National Science Foundation and US National Institute of Health (grants to Brian Wandell, Stanford University, NSF BCS1228397 and NIH NEI EY015000), Indiana University College OF Arts and Sciences Startup funds and Indiana Clinical and Translational Institute CTSI (GLUE Grant; supported by NIH grants ULTTR001108, TR001106, TR001107). Ariel Rokem was supported by the National Institute of Health (NRSA F32-EY022294), the Alfred P Sloan Foundation and the Gordon & Betty Moore Foundation (UW eScience Institute Data Science Environment). The authors wish to thank all the participants for taking part in this study. We are grateful to Brian Wandell for generous support and advice and for hosting Dr. Ajina while undertaking this work.

## Additional information

### Funding

| Funder | Grant reference | Author |
|---|---|---|
| Wellcome Trust | Clinical Research Fellowship | Sara Ajina |
| Royal Society | University Research Fellowship | Holly Bridge |

| Funder | Grant reference | Author |
|--------|-----------------|--------|
| National Science Foundation (NSF) | NSF BCS1228397 to Brian Wandell | Franco Pestilli |
| National Institute for Health Research (NIHR) | Biomedical Research Centre | Christopher Kennard, Sara Ajina |
| National Institutes of Health (NIH) | NIH ULT TR001108 | Franco Pestilli, Ariel Rokem |
| National Eye Institute (NEI) | EY0155000 to Brian Wandell | Franco Pestilli, Ariel Rokem |
| National Research Service Award (NRSA) | F32-EY022294 | Franco Pestilli, Ariel Rokem |

The funders had no role in study design, data collection and interpretation, or the decision to submit the work for publication.

## Author contributions

SA, Conception and design, Acquisition of data, Analysis and interpretation of data, Drafting or revising the article; FP, AR, Analysis and interpretation of data, Drafting or revising the article; CK, Conception and design, Drafting or revising the article; HB, Conceived and designed the experiment. Helped with analysis and interpretation of data. Revised and updated article, Conception and design, Analysis and interpretation of data, Drafting or revising the article

## Ethics

Human subjects: Ethical approval was provided by the Oxfordshire Research Ethics Committee B (Ref B08/H0605/156). All participants gave informed, written consent.

# Additional files

## Supplementary files

• Supplementary file 1. Clinical characteristics of patients.

• Supplementary file 2. Number of uncleaned fascicles for the three pathways of interest in patients and control participants: (1) Ipsilateral LGN and hMT+ (2) hMT+ bilaterally via the corpus callosum (3) Ipsilateral SC and hMT+. Results are shown separately for the intact and damaged 'ipsi-lesion' hemispheres (right and left for control participants).

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
