## [Decision Letter]

Thank you for submitting your work entitled “Human blindsight is mediated by an intact geniculo-extrastriate pathway” for peer review at *eLife*. Your submission has been favorably evaluated by Eve Marder (Senior Editor) and two reviewers, one of whom is a member of our Board of Reviewing Editors.

The authors performed diffusion weighted MRI imaging in 17 patients with cortical blindness and 9 matched controls. Patients were categorized as displaying blindsight (PB, N = 12) or not (PN, N = 5) based on above chance detection of Gabor patches presented in their affected visual field. The results showed that ipsilesional LGN-hMT + tracts could be reconstructed in all subjects, except 1 PN patient. Investigations of the microstructure of the tract by means of FA and MD revealed normal values for the PBs and abnormal values for the PNs. In addition, the authors investigated 2 alternative projections, i.e. ipsilesional SC-hMT+ and transcallosal hMT+. These tracts could not be identified in all PB patients and all controls, whereas they were present in some PN patients. The authors interpret the results as evidence that an intact LGN-hMT + pathway is both a necessary and sufficient condition for blindsight.

Essential Revisions:

Interpretation:

1) Considering the limited number of tracts investigated (i.e. 3), any claims on the “crucial”, “necessary” or “sufficient” nature of the results, seems a bit over-ambitious. A more provisional interpretation would be more in line with the a priori constraints that the authors choose to apply in the analysis.

2) Can the authors provide any evidence that the LGN-hMT + connectivity results are not explained by grey matter volume of the hMT+? In other words, could the connectivity results secondary to structural integrity of the grey matter of hMT+?

3) There is an additional alternative tract that the authors do not investigate, namely a pulvinar-hMT + tract (e.g. Leh et al., 2008; Wong et al., 2009). Can the authors please comment and address this.

Study Execution and Statistical Analysis:

1) Would the authors say a little more about how the controls were chosen? What criteria were they matched on relative to the blindsight patients? The definition of blindsight depends only on the responds at the 100% contrast level. If this was the plan prospectively in the design if the study then what was the purpose of testing at the other 4 contrast levels? How many correct responses did the patient have to have to be considered blindsight positive?

2) The reviewers did not find a formal statistical assessment showing that the performance difference between the two groups was statistically significant or that the performance in the blindsight positive group was significantly greater than chance, whereas the blindsight negative group was at chance. Moreover, we did not see an assessment that showed that performance between the blindsight positive and the control group was similar. We suspect that the differences between the two groups are even stronger than the authors indicate. It would be more efficient to analyze the data as a logistic regression within subject in which detection performance is measured as a function of contrast. That is use the model Log[(1-p)/p)] = a + b*percent contrast which can easily be fit by using standard statistical software.

For each subject all of the trials go into estimating just two parameters per subject which is far more efficient than estimating response probabilities at different contrast levels, assuming that performance at each contrast level within a subject is unrelated. At a given contrast level for each subject in the blindsight positive group, if the lower bound of the 95% confidence bound is above 0.5 then we can infer that that subject's performance is greater than chance. Similarly, for each subject in the blindsight negative group the 95% confidence intervals at each contrast level should include 0.5.

There should be at least one other formal comparison. There should be an assessment that shows that the performances of the blindsight positive group and the controls are indistinguishable. These are easy to do if a Bayesian hierarchical analysis is used to pool information across the two groups. It suffices to show that confidence interval (posterior credibility intervals) for the difference between the population logistic curves for the two groups include zero.

3) A lot of quantitative comparisons are reported throughout the Results. However, formal statistical testing only appears the subsections “Different microstructure in the geniculate-hMT + tract between blindsight positive and negative patients” (paragraph five),“Superior colliculus tracts” (paragraph three), and “Lesion size and location” (paragraph one). Moreover, these tests are not described in the Methods. Can the authors give a bit better discussion in the Methods about what statistical paradigm they are using to make their inferences? Moreover, formal testing of group differences in both macro- and microstructural parameters of the 3 (or 4 if the suggested one is included) projections of interest could strengthen the claims of the authors, as well as correlation analyses between behavioral and imaging data. This is one of the largest collections of V1 lesion patients. Therefore the data should be analyzed in greater detail.

---

## [Author Response]

Essential Revisions:

Interpretation:

*1) Considering the limited number of tracts investigated (i.e. 3), any claims on the “crucial”, “necessary” or “sufficient” nature of the results, seems a bit over-ambitious. A more provisional interpretation would be more in line with the* a priori *constraints that the authors choose to apply in the analysis.*

This is a fair point, and the language has been toned down to be less absolute in several locations:

“In contrast, the two alternative candidate tracts showed variable predominance in both patient groups and therefore seem unlikely to underlie blindsight function.” (Introduction, paragraph five)

“This has two important implications that support geniculate-extrastriate connections in blindsight […] it is possible that another, unexplored, pathway could underlie blindsight function in these patients.” (subsection “A direct geniculate pathway consistently supporting blindsight function”)

Finally, the title of the subsection “Intact collicular or interhemispheric pathways are unlikely to underlie blindsight” has been changed.

2) Can the authors provide any evidence that the LGN-hMT + connectivity results are not explained by grey matter volume of the hMT+? In other words, could the connectivity results secondary to structural integrity of the grey matter of hMT+?

This is a good question, and in order to address this we measured the grey matter volume within the hMT+ masks and performed a 2-way ANOVA to determine whether there was an effect of either blindsight status (positive or negative) or lesion side (ipsi-lesion or contra-lesion).

The following paragraph has been added to the Results, in the subsection “Different white matter microstructure in the geniculate-hMT + tract between blindsight positive and negative patients”:

“To ensure that any differences in tract microstructure between blindsight positive and negative patients […] are unlikely to have affected the results significantly.”

3) There is an additional alternative tract that the authors do not investigate, namely a pulvinar-hMT + tract (e.g. Leh et al., 2008; Wong et al., 2009). Can the authors please comment and address this.

This is a very good point, and one that we have considered carefully. Although at this point we have not performed the tractography, we have explained our reasons for not doing this in the main text (please see subsection “Pulvinar pathways to hMT+”).

Study Execution and Statistical Analysis:

1) Would the authors say a little more about how the controls were chosen? What criteria were they matched on relative to the blindsight patients? The definition of blindsight depends only on the responds at the 100% contrast level. If this was the plan prospectively in the design if the study then what was the purpose of testing at the other 4 contrast levels? How many correct responses did the patient have to have to be considered blindsight positive?

To elaborate on how control subjects were selected, the following sentences have been incorporated into the Methods:

“Thus, control and patients were matched by age and sex at the time of testing. Controls all had normal or corrected-to-normal visual acuity and no history of neurological disease.”

The definition of blindsight used the same criteria as in previous work (Ajina et al., 2015). Participants either achieved an average score across all contrast levels, or a score for just the highest contrast stimuli (i.e. 100% contrast) that was significantly greater than chance. In both cases, statistical significance needed to surpass a threshold of p < 0.01 using a cumulative binomial distribution. For each contrast level, trials were repeated 20 times. Therefore participants either had to score at least 16 correct responses for stimuli of 100% contrast, or at least 63/100 correct responses across all contrast levels. This meant that patients could be labelled as blindsight positive if they demonstrated a consistent performance above chance, even if this was not just for the highest contrast level. Whilst performance in blindsight is often considered optimal for high luminance stimuli, studies have also identified patients with superior contrast sensitivity to normal in the blind hemifield (Trevethan et al., 2007).

An overview of this procedure is in the Methods section.

2) The reviewers did not find a formal statistical assessment showing that the performance difference between the two groups was statistically significant or that the performance in the blindsight positive group was significantly greater than chance, whereas the blindsight negative group was at chance. Moreover, we did not see an assessment that showed that performance between the blindsight positive and the control group was similar. We suspect that the differences between the two groups are even stronger than the authors indicate.

The reviewers are correct that currently there is no analysis indicating that there is a significant difference between the blindsight positive and negative groups. Each individual patient has to score p < 0.05 above zero (as detailed above) to be classified at blindsight positive. For this reason, there is a highly significant difference in performance at the 100% contrast level between the two groups (t = 10.62; p < 0.0001), but it also means that this statistic is not valid since the groups have already been divided using this metric and we would therefore prefer not to use this statistic in the manuscript.

With regard to comparison with control participants, there is no comparable behavioural data in the sighted subjects as the contrast task is too easy for them to perform as they would score 100% at all contrasts. Indeed, when the sighted hemifield of patients was tested as a comparison, even the lowest contrast stimulus of 1% was associated with 100% detection in all participants.

A comment to reinforce the point about the control subjects and sighted hemisphere has been added to the Methods, in the subsection “Psychophysics”.

“Behavioural testing of control participants and the sighted hemisphere of patients was not possible, since the contrast task is too easy resulting in 100% detection of even the 1% control stimulus.”

*It would be more efficient to analyze the data as a logistic regression within subject in which detection performance is measured as a function of contrast. That is use the model Log[(1-p)/p)] = a + b*percent contrast which* can *easily be fit by using standard statistical software.*

*For each subject all of the trials go into estimating just two parameters per subject which is far more efficient than estimating response probabilities at different contrast levels, assuming that performance at each contrast level within a subject is unrelated. At a given contrast level for each subject in the blindsight positive group, if the lower bound of the 95% confidence bound is above 0.5 then we* can *infer that that subject's performance is greater than chance. Similarly, for each subject in the blindsight negative group the 95% confidence intervals at each contrast level should include 0.5.*

There should be at least one other formal comparison. There should be an assessment that shows that the performances of the blindsight positive group and the controls are indistinguishable. These are easy to do if a Bayesian hierarchical analysis is used to pool information across the two groups. It suffices to show that confidence interval (posterior credibility intervals) for the difference between the population logistic curves for the two groups include zero.

As mentioned above, it is not possible to do the Bayesian hierarchical analysis to compare sighted controls (or sighted hemifield) with the blindsight patients as performance is at ceiling (100%) for all contrasts. Therefore the two groups will be different. The other suggested analysis has been implemented for the patient groups, but while the logistic function fits the data from patients that are blindsight positive well (average coefficient of determination larger than 80%), the function is not well-fit to the blindsight negative participants (average coefficient of determination smaller than 20%). For that reason, in addition to the circularity of the comparison, mentioned above, we opted not to include this analysis in the manuscript. We have, however, included two additional cross-validation analyses to confirm the classification into these two groups, both of which use all performance data for each participant. These are described in the subsection “Behavioural measurements of blindsight” (and additional details are in the Methods):

“Classification of participants as either “blindsight positive” or “blindsight negative” was further validated using cross-validation. […] Each participant was then assigned to one of the two classes based on their similarity to the centroid of each of these distributions. Both classification algorithms agree with the distinction based on performance in 100% contrast in all cases.”

3) A lot of quantitative comparisons are reported throughout the Results. However, formal statistical testing only appears the subsections “Different microstructure in the geniculate-hMT + tract between blindsight positive and negative patients” (paragraph five),“Superior colliculus tracts” (paragraph three), and “Lesion size and location” (paragraph one). Moreover, these tests are not described in the Methods. Can the authors give a bit better discussion in the Methods about what statistical paradigm they are using to make their inferences? Moreover, formal testing of group differences in both macro- and microstructural parameters of the 3 (or 4 if the suggested one is included) projections of interest could strengthen the claims of the authors, as well as correlation analyses between behavioral and imaging data. This is one of the largest collections of V1 lesion patients. Therefore the data should be analyzed in greater detail.

Thank you for the comments about the data. A section has been added to the Methods to explain the statistical tests employed, particularly since we have added additional statistical tests as requested.

Statistical analyses (ANOVA) have been added to determine the difference in microstructural properties between blindsight positive and negative groups. Unfortunately, since only 2 of the blindsight negative patients showed a tract between hMT+ in the two hemispheres, it was not possible to perform a significance test. The following passages have been added to the manuscript:

“The differences between blindsight patients and lesion side can be illustrated using a two-way ANOVA of the FA values within the geniculate-hMT + tract […] significant effect of lesion side (F = 35.7; p < 0.00005) and interaction (F = 12.4; p < 0.005).” (in the subsection “Different white matter microstructure in the geniculate-hMT + tract between blindsight positive and negative patients”)

“Although the effect of blindsight status on mean FA within this distal portion was not […] as were the effect of lesion side (F = 31.4; p < 0.0001) and the interaction (F = 15.6; p = 0.001).” (in the subsection “Although the effect of blindsight status on mean FA”)

“Statistical comparison of these values is complicated […] ipsilesional hemisphere indicated no significant difference in either measure between the two groups (FA: t = 0.9; p = 0.4; d.f. = 9; MD: t = 0.8; p = 0.45; d.f. = 9).” (in the subsection “Superior colliculus tracts”)

Following the suggestion of the reviewers, we also performed statistical testing of the correlation between blindsight performance and tract microstructure. We have added a section to the Results laying out these statistics, and an additional figure showing the data (please see “Relationship between blindsight performance and tract microstructure”).

Interpretation of these results have also been added to the Discussion (in a subsection also entitled “Relationship between blindsight performance and tract microstructure”):

“Only the LGN-hMT + tract showed a correlation of behavioural performance with MD and FA that was in the correct direction: improved performance correlated positively with FA and negatively with MD. However, neither of these correlations was significant. Although the current study provides a relatively large participant group compared to other hemianopia work, the considerable variability in values for tract microstructure means that there may not be sufficient power to find differences, particularly for smaller tracts.”